# Interleukin-6 in Hepatocellular Carcinoma: A Dualistic Point of View

**DOI:** 10.3390/biomedicines11102623

**Published:** 2023-09-24

**Authors:** Iuliana Nenu, Teodora Maria Toadere, Ioan Topor, Andra Țichindeleanu, Daniela Andreea Bondor, Șerban Ellias Trella, Zeno Sparchez, Gabriela Adriana Filip

**Affiliations:** 1Department of Physiology, “Iuliu Hațieganu” University of Medicine and Pharmacy, 400006 Cluj-Napoca, Romania; nenu.iuliana@elearn.umfcluj.ro (I.N.); toadere.teodora.maria@elearn.umfcluj.ro (T.M.T.); tichindeleanu.andra@elearn.umfcluj.ro (A.Ț.); bondor.daniela.andre@elearn.umfcluj.ro (D.A.B.); trella.serban.ellias@elearn.umfcluj.ro (Ș.E.T.); gabriela.filip@umfcluj.ro (G.A.F.); 2Department of Gastroenterology, “Prof. Dr. O. Fodor” Regional Institute of Gastroenterology and Hepatology, 400162 Cluj-Napoca, Romania; zsparchez@gmail.com; 3Department of Internal Medicine, “Iuliu Hațieganu” University of Medicine and Pharmacy, 400162 Cluj-Napoca, Romania

**Keywords:** hepatocellular carcinoma, interleukin 6, liver immunology, HCC therapies

## Abstract

Hepatocellular Carcinoma (HCC) is a pressing health concern, demanding a deep understanding of various mediators’ roles in its development for therapeutic progress. Notably, interleukin-6 (IL-6) has taken center stage in investigations due to its intricate and context-dependent functions. This review delves into the dual nature of IL-6 in HCC, exploring its seemingly contradictory roles as both a promoter and an inhibitor of disease progression. We dissect the pro-tumorigenic effects of IL-6, including its impact on tumor growth, angiogenesis, and metastasis. Concurrently, we examine its anti-tumorigenic attributes, such as its role in immune response activation, cellular senescence induction, and tumor surveillance. Through a comprehensive exploration of the intricate interactions between IL-6 and the tumor microenvironment, this review highlights the need for a nuanced comprehension of IL-6 signaling in HCC. It underscores the importance of tailored therapeutic strategies that consider the dynamic stages and diverse surroundings within the tumor microenvironment. Future research directions aimed at unraveling the multifaceted mechanisms of IL-6 in HCC hold promise for developing more effective treatment strategies and improving patient outcomes.

## 1. Introduction

Interleukin-6 (IL-6) is a pro-inflammatory cytokine that plays a crucial role in the development and progression of hepatocellular carcinoma (HCC), the most common type of liver cancer. IL-6 is produced by various cell types, including immune cells, stromal cells, and cancer cells themselves. 

In hepatocellular carcinoma (HCC), IL-6 exhibits a dual role, with its effects being contingent upon its context. While IL-6 predominantly promotes tumor growth and advancement, it also exhibits a paradoxical impact in specific scenarios, showcasing the contrasting influences of this cytokine. This dichotomy often corresponds to different aspects of the tumor microenvironment and the tumor’s developmental stage. On the one hand, IL-6 is instrumental in bolstering the proliferation and survival of HCC cells, achieved through the activation of the JAK-STAT3 signaling pathway. This activation leads to the elevation of genes responsible for cell proliferation and the inhibition of apoptosis, thus contributing to unregulated tumor cell expansion. When it comes to angiogenesis, IL-6 propels the development of fresh blood vessels within the tumor (angiogenesis), furnishing crucial nutrients and oxygen to sustain tumor growth and enlargement. Moreover, IL-6 is correlated with heightened metastatic potential in HCC. It facilitates the process of epithelial–mesenchymal transition (EMT), empowering cancer cells to acquire invasive attributes and disseminate to distant organs [1,2].

On the other hand, in certain situations, IL-6 can stimulate an anti-tumor immune response. It can promote the activation and recruitment of immune cells such as cytotoxic T cells and natural killer cells, which play a crucial role in recognizing and eliminating cancer cells. IL-6 is involved in immune surveillance, which is the process of the immune system recognizing and controlling cancer cells. It can trigger the immune system to detect and eliminate pre-cancerous cells [3,4].

The dual role of IL-6 in HCC can be partly explained by the heterogeneity of the tumor microenvironment and the complex interplay between cancer cells, immune cells, and stromal cells. Additionally, the stage of the tumor, the level of IL-6 expression, and the presence of other cytokines and signaling molecules can influence its effects. Therefore, the goal of this review is to highlight the importance of custom therapeutic strategies for HCC patients, taking into consideration the dynamic nature of the interactions between Il-6 and the tumor microenvironment.

## 2. Interleukin-6—In the Maze of Physiological and Pathological

Interleukin-6 (IL-6) is a 184 amino acid glycosylated protein formed by four-helix bundles (A-D) arranged as a ribbon-shaped molecule. A diverse range of cellular sources (monocytes, fibroblasts, tumor, endothelial, mesenchymal, and T-cells) synthesizes this multifunctional cytokine [5,6]. Initially identified as a Bcell stimulatory factor, it sizes 21–26 kDa and partakes in the interleukin-6 cytokine family along with interleukin-11 (IL-11), ciliary neurotrophic factor (CNTF), leukemia inhibitory factor (LIF), oncostatin M (OSM), cardiotrophin-like cytokine (CLC), cardiotrophin-1 (CT-1), novel neutrophin-1 (NNT-1), and interleukin-27 (IL-27). IL-6 distinguishes itself from the other family members by its unique signaling mechanisms, which will subsequently be elucidated in this paper. Physiological blood levels of interleukin-6 (IL-6) typically range from 1 to 5 pg/mL but can increase several thousand-fold during inflammatory states. In lethal septic conditions, IL-6 levels can reach up to several mg/mL [3,5,7,8].

IL-6 is involved in inflammation, immune response, hematopoiesis, metabolism, embryonic development, and memory consolidation by controlling its targeted cells’ differentiation, proliferation, migration, and apoptosis. It stimulates B and T cells, induces hepatic acute phase proteins (C reactive protein, serum amyloid A, fibrinogen, hepcidin) and targets bone marrow and synovial and dermal fibroblasts [3,8,9]. Consequently, dysregulations involving the participation of IL-6 hold substantial clinical significance.

Regarding extracellular signaling pathways, IL-6 can exert classic signaling (Figure 1A) or trans-signaling (Figure 1B). The classical pathway implies the binding of IL-6 with the membrane-bound IL-6 receptor (mIL-6R), a ligand-binding glycoprotein called CD126. mIL-6R is a transmembrane α-chain (80 kDa) consisting of three domains (D1, D2, and D3). The Ig domain, referred to as D1, is connected to the domain responsible for binding cytokines (CBD). Utilizing the CBD, IL-6 attaches to the co-receptor glycoprotein 130 (gp130), forming a hexameric arrangement [8]. The following point is crucial to highlight: mIL-6R lacks signal-transduction capacity, whereas gp-130 possesses signal-transduction capabilities [10]. This matter signifies that only gp130 can trigger internal signaling sequences. IL-6 itself does not exhibit any binding affinity toward gp130; therefore, mIL-6R is imperative for the initial binding of IL-6 and the subsequent activation of gp130. Neither the cytoplasmic nor the transmembrane segments of IL-6R hold signaling competence [5]. mIL-6R is expressed on a few cells (hepatocytes, neutrophils, monocytes, T-cells), while gp130 is expressed on all human cells. This hexameric complex formed by IL-6, mIL-6R, and gp-130 activates Janus kinase (JAK), activating different signaling routes. One signaling route represents the self-tyrosine phosphorylation of JAK, which then induces the dimerization of signal transducer and transcription-3 (STAT3). In an alternate signaling pathway, JAK triggers the Ras/Raf pathway, leading to an elevated level of hyperphosphorylation of mitogen-activated protein kinases (MAPK). This results in an augmentation of serine and threonine kinase activity within MAPK. Additionally, JAK can phosphorylate and activate phosphoinositol-3 kinase (PI3K). Subsequently, PI3K phosphorylates specific phosphatidylinositides, converting them into phosphatidylinositol-4,5-bisphosphate (PIP2) and phosphatidylinositol-3,4,5-trisphosphate (PIP3). PIP3 serves as the activator for the protein kinase B (PKB)/Akt pathway, contributing to activating the nuclear factor kappa light chain enhancer of activated B cells (NF-kB). Consequently, this classical signaling pathway governs essential physiological functions, including initiating acute phase proteins (APP) within the liver, impacting neutrophils, and influencing acquired immunity. Nevertheless, it also plays a role in pathological conditions such as cancer by activating the STAT3, MAPK, and PKB/Akt pathways, and NF-kB [6,9,11,12]. This particular way of signaling is also associated with the anti-inflammatory roles that IL-6 holds.

The conventional IL-6R signaling pathway exhibits limitations due to its reliance on the presence of mIL-6R to confer responsiveness to IL-6 within a cell. This is where the concept of trans-IL-6 signaling comes into play. The soluble IL-6 receptor (sIL-6R), which ranges from 50 to 55 kDa, constitutes the extracellular component of the transmembrane IL-6R. It arises either through proteolytic cleavage or by translating an alternatively spliced mRNA, wherein the exon responsible for encoding the transmembrane segment is omitted [7]. The latter accounts only for a minor proportion (1–10%) of sIL-6R, the vast majority (90–99%) being produced by proteolytic cleavage of the IL-6R by two Zn2+-dependent proteases: ADAM10 and ADAM17, both members of the A Disintegrin and Metalloproteinase (ADAM) family. Consequently, sIL-6R gains the ability to freely traverse the organism, dispersing it into various bodily fluids where it can engage its ligand, IL-6. Within the serum of healthy individuals, a concentration of approximately 25–50 ng/mL of sIL-6R is typically detected. Upon interaction with gp130, the complex formed by IL-6 and sIL-6R kickstarts identical intracellular signaling pathways, as outlined earlier. The difference lies in their effect on cells that express gp130 but lack IL-6R, effectively stimulating this subset of cells [6,9]. Trans IL-6 signaling is linked with pro-inflammatory outcomes.

In a pathological context, IL-6 plays a significant role in cancer development. Elevated levels of IL-6 have been observed in various tumor types, including colon, ovarian, pancreatic, lung, hepatocellular, breast, prostate, and multiple myeloma [13,14,15,16,17,18,19,20]. The three intracellular signaling pathways (JAK/STAT3, Ras/Raf/MAPK, PI3K/PKB/Akt) initiated by IL-6 cause cell proliferation, differentiation, apoptosis, and metastasis [21]. IL-6 regulates CD4+ T-cells by inhibiting regulatory T cell (T_reg_) differentiation and stimulating the differentiation of T helper 17 cells (Th17, pro-inflammatory cells that produce IL-17). 

These disruptions lead to an immune system imbalance, resulting in the secretion of IL-17, IL-6, and tumor necrosis factor (TNF-α). Through activating JAK, IL-6 also triggers the gene expression of factors such as Bcl-2, Bcl-xl, Survivin, Mcl-1, and XIAP, all associated with apoptotic cell processes [22]. The PI3K/Akt pathway also contributes to NF-kB activation, as mentioned before. This promotes enhanced cell survival and governs tumor invasion. Furthermore, IL-6 exerts its effects on synovial fibroblasts and lymphoid follicles, inducing elevated production of vascular endothelial growth factor (VEGF). It can result in heightened vascular permeability, leading to vascular leakage and joint destruction due to pannus formation. Collectively, these factors, including the excessive expression of IL-17, IL-6, TNF-α, altered gene expression, and vascular leakage, contribute to a cancer-favorable environment [6].

## 3. HCC—New Pieces to the Puzzle

Hepatocellular carcinoma stands as the fourth leading contributor to cancer-related fatalities globally. It constitutes 90% of primary liver cancers, and the mortality rates consistently rise by 2 to 3% annually. This escalation is primarily attributed to frequently delayed diagnosis and the absence of curative treatments for the advanced stages of the disease [23].

Hepatitis B virus (HBV) and hepatitis C virus (HCV) infections, excessive alcohol consumption (which, although not mutagenic, heightens cirrhosis risk), and the ingestion of aflatoxin B1 are key factors contributing to the development of hepatocellular carcinoma (HCC) (Figure 2) [24]. Moreover, the increasing incidence rate of hepatocellular carcinoma is becoming progressively more associated with non-alcoholic fatty liver disease (NAFLD). NAFLD encompasses a range of chronic liver conditions characterized by the accumulation of triglycerides within hepatocytes, exceeding 5% of their cytoplasmic content. Importantly, this occurs in the absence of significant alcohol consumption and is notably correlated with metabolic disorders such as obesity, insulin resistance, and type 2 diabetes. NASH (non-alcoholic steatohepatitis) is increasing, and the mortality from HCC is predicted to reach 1 million deaths annually by 2030 [25,26,27]. In addition, smoking tobacco increases HCC by accelerating liver fibrosis, and long-term usage is strongly associated with it [28,29] (55% increased risk for HCC was demonstrated in a cohort study, the “Liver Cancer Pooling Project” [30]). Smoking brings hepatocarcinogens (e.g., 4-aminobiphenyl and acetylaminofluorene) into the body, which increases the production of proinflammatory cytokines and induces oxidative stress, triggering lipid peroxidation [31].

Early detection of hepatocellular carcinoma (HCC) traditionally involves surveillance through techniques such as ultrasonography (US) and serological measurements of alpha-fetoprotein (AFP). In the case of lesions smaller than 1 cm, regular screening is advised, typically involving follow-up US examinations at intervals of 3 to 6 months, as outlined in diverse guidelines [32,33]. In lesions ≥1 cm, magnetic resonance imaging (MRI), computed tomography (CT), and other cross-sectional imaging techniques provide a definitive diagnosis. Moreover, although many scientists are now seeking new biomarkers due to the controversy regarding the utility of AFP, it remains the most universally used biomarker for HCC [34]. It has been confirmed that a persistently increased AFP level, which has been proven to be associated with an aggressive histological morphology (vascular invasion (poorly differentiated) and satellitosis), is a hazardous factor for HCC [35].

Alpha-fetoprotein (AFP) presents in three distinct glycoforms, each exhibiting varying affinities for binding to the lectin Lens culinaris agglutinin (LCA). These glycoforms are classified as AFP-L1 (non-binding fraction), AFP-L2 (weak binding fraction), and AFP-L3 (binding fraction). Notably, AFP-L1 experiences elevation in cases of chronic hepatitis and liver cirrhosis, whereas AFP-L3 demonstrates specific elevation solely in hepatocellular carcinoma (HCC). AFP-L3 is exclusively produced by cancer cells, rendering it a more targeted biomarker for HCC [36,37]. However, AFP-L3 is not applicable for HCC diagnosis when total AFP levels are below 20 ng/mL as it remains undetectable in such cases. When the total AFP concentration is less than 20 ng/mL, AFP-L3 is irrelevant for HCC detection as it remains untraceable [38].

Des-γ-carboxyprothrombin (DCP) is an abnormal prothrombin molecule that is increased in HCC. DCP loses its normal prothrombin function but may take on an essential role in promoting malignant proliferation in HCC, and many studies have shown that the level of serum DCP in patients with benign and malignant liver diseases deviates significantly from physiological [39]. Glypican-3 (GPC3) belongs to the glypican family of heparan sulfate proteoglycans and is involved in cell proliferation, survival, and tumor suppression. Usually, it is absent in healthy and non-malignant hepatocytes but appears to function differently in diverse cancers. GPC3 is downregulated in breast cancer, ovarian cancer, and lung adenocarcinoma, and it is upregulated in HCC [40,41]. A study by Nakatsura et al. [42] found that GPC3 was detected in 40% of HCC patients and 33% of HCC patients seronegative for both AFP and DCP.

Osteopontin (OPN), also known as the transformation-related protein phosphatase, is an integrin-binding glycophosphoprotein overexpressed in many different types of malignancies, including lung, breast, and colon cancer. The protein has been found to play a role in many physiological cellular functions, including migration, invasion, and metastasis [43]. One of its more critical roles has been suggested to be in the metastatic potential of various cancers [44]. Commonly, OPN is expressed in bile duct epithelium, stellate cells, and Kupffer cells but not in hepatocytes [45]. However, elevated expression of serum OPN has been reported in HCC patients compared to normal liver patients or those with liver cirrhosis or chronic hepatitis [46,47]. For OPN, the AUC for discriminating between early-stage HCC (BCLC stage A) and cirrhosis was 0.73 [38]. 

Golgi protein-73 (GP73) is a type II Golgi-specific membrane protein typically expressed in the epithelial cells of various human tissue types but not hepatocytes. However, GP73 is detected in the serum of patients with liver disease, particularly HCC [48]. A case-control study by Mao Y et al. [49] demonstrated that serum GP73 in patients with HCC was significantly elevated than compared to those in healthy adults and hepatitis B virus (HBV) carriers without hepatic disease. The combination of GP73 and AFP increased the sensitivity and specificity to 89.2% (95% CI: 86.7–91.5%) and 85.2% (95% CI: 83.4–86.4%), respectively, with an AUC of 0.96 [50] (Figure 3).

Early and accurate diagnosis of HCC patients is critical for patient prognosis. Several studies have successfully identified promising biomarkers for the diagnosis of HCC. However, current studies suggest that a single biomarker alone may not have the best sensitivity and specificity for detecting HCC, especially for detecting early-stage HCC [53]. Biomarker combinations can further increase the predictive performance for HCC detection [54]. Moreover, it has been reported that combining several biomarkers improves the early diagnosis rate [50]. Adding clinical variables (such as age and gender) into the model, it is worth noting that the GALAD score, which is the combination of clinical factors (gender, age) and biomarkers (AFP, AFP-L3, and Des-carboxyprothrombin), has been validated to improve the performance of discerning between HCC and cirrhosis [55].

A study by Gosain et al. aimed to establish the role of IL-6 as a prognostic biomarker in patients with both hepatocellular carcinoma (HCC) and biliary cancer. The authors assessed how IL-6 impacts two crucial factors affecting quality of life: pain scores and performance status.

In order to minimize potential confounding variables, the study researchers selected a control group comprising 91 individuals. These controls were carefully matched in terms of age, gender, and BMI, as these factors are widely recognized for their substantial impact on IL-6 levels [56,57,58,59,60,61]. The research team collected blood samples from the patients, and serum specimens were determined using a commercially available enzyme-linked immunosorbent assay (ELISA) kit. IL-6 levels were notably higher in hepatobiliary cancer patients when compared to the healthy control group.

In conclusion, elevated levels of IL-6 were closely associated with poorer prognoses, increased pain scores, and deteriorated performance statuses in the patients [62]. 

A retrospective study by Yang et al. aimed to assess the utility of IL-6 promoter methylation levels as a noninvasive diagnostic biomarker for liver cancer. The study involved 165 patients diagnosed with HCC associated with hepatitis B virus, 198 patients with chronic hepatitis B, and 31 healthy individuals serving as controls. The research employed Methylight methodology to detect methylation levels in the IL-6 promoter region within peripheral blood mononuclear cells (PBMCs) [63].

In conclusion, the study revealed that in HBV-associated HCC, the methylation levels of the IL-6 promoter were notably lower than those observed in patients with chronic hepatitis B. Conversely, the IL-6 mRNA levels were significantly elevated in HBV-associated HCC compared to both CHB patients and healthy individuals [63].

## 4. Is There a Role of IL-6 in HCC Biology?

IL-6 plays diverse roles, acting as a hepatocyte stimulatory factor to trigger acute phase reactions. However, its presence also heightens susceptibility to various human ailments, including autoimmune diseases, chronic inflammatory conditions, and multiple forms of cancer. IL-6 has been identified as a cytokine that is significantly abundant within the tumor microenvironment (TME) across a spectrum of cancer types. These include head and neck squamous cell carcinoma (HNSCC), pancreatic cancer, non-small-cell lung cancer, breast cancer, ovarian cancer, and melanoma, as substantiated by previous studies [18,64,65,66,67,68,69,70,71,72,73]. Generally, the overexpression of IL-6 is associated with poor prognosis and a low survival rate in patients with breast cancer [74]. Beyond its recognized role in tumorigenesis, IL-6 also plays a pivotal role in orchestrating the intricate sequence of events that precedes the establishment of secondary tumors, commonly referred to as metastasis [75]. Notably, elevated serum IL-6 levels have been detected in patients with primary liver cancer. 

IL-6 engages three significant pathways (STAT3, MAPK, and PI3K). Among these, the STAT3 pathway holds particular significance, being classified as an oncogene and consistently activated in multiple human malignancies. Ordinarily, IL-6 functions to confer hepatoprotection against apoptosis stemming from viral infections or exposure to chemicals, primarily through the STAT3 pathway [76,77,78]. 

The roles of IL-6 in HCC are emphasized by the effects of the IL-6/STAT3 signaling pathway [79]. The hyperactivation of STAT3 holds a significant role in the formation of inflammatory tumors within the microenvironment, facilitating both uncontrolled proliferation of cancer cells and metastasis [80]. In particular, the genes responsible for promoting cancer cell proliferation, including Ras, Src, and cyclin D1, serve as direct targets of STAT3 [81].

Tumor-associated macrophages (TAMs) promote tumor progression in liver tissue microenvironments by releasing IL-6, which, in turn, activates the IL-6/STAT3 signaling pathway in nearby HCC stem cells [82]. A study by Zheng et al. [83] sheds light on the hyperactivation of the IL-6/STAT3 signaling pathway in HCC cells which could upregulate the tissue inhibitor of metalloproteinases-1 (TIMP-1), thus prompting the conversion of normal liver fibroblasts (LFs) into carcinoma-associated fibroblasts (CAFs), ultimately driving the initiation of liver cancer [2].

The induction of apoptosis in HCC cells primarily hinges on two mechanisms: the upregulation of anti-apoptotic factors and the promotion of survival signals. Following STAT3 activation mediated by IL-6, there is an important increase in the expression of anti-apoptotic proteins such as Bcl-xL, Bcl-2, survivin, and P53, among others. This upregulation plays a crucial role in safeguarding HCC cells against apoptosis [83,84,85,86]. In particular, Bcl-2 stands out as a vital protein in fostering the survival of tumor cells.

The critical determinant of apoptosis is the delicate balance between pro-apoptotic and anti-apoptotic proteins. Activation of the IL-6/STAT3 signaling pathway has the potential to shift this balance by increasing the proportion of pro-apoptotic factors relative to anti-apoptotic factors, and heightened IL-6 levels are likely to contribute to this modification [87].

Additionally, STAT3 phosphorylation can directly bind to the promoter region of the survivin gene, leading to an upregulation of survivin expression and thereby promoting the survival of tumor cells. On the other hand, inhibiting STAT3 activity can result in the downregulation of survivin gene expression, facilitating the induction of apoptosis in liver cancer cells [88]. These findings demonstrate that the activation of the IL-6/STAT3 signaling pathway has the capacity to boost the expression of survival-related proteins, which, in turn, act to impede apoptosis in HCC cells.

A crucial role in tumor invasion and metastasis is vascular endothelial growth factor (VEGF), which promotes vascular endothelial cell growth and tumor neoangiogenesis. The expression of VEGF is more elevated in liver tumor tissue than in cirrhosis and normal liver tissue [89]. A significant factor which can induce the secretion and expression of VEGF in tumor tissue is hypoxia through hypoxia inducible factor 1 (HIF-1). IL-6 binds with IL-6R to induce the activation of STAT3 and activated STAT3 binds to the promoter region of the VEGF gene to increase transcription, promoting the tumor angiogenesis.

HCCs that are associated with ASH (alcoholic steatohepatitis) or NASH (non-alcoholic steatohepatitis) have a similar molecular key: the IL-6/JAK/STAT signaling pathway [90] activated by the interleukin-6 family of cytokines. This family’s hallmark is using the signal-transducing β-receptor gp130 [91].

Estrogen inhibits the release of IL-6 from Kupffer cells, reducing the risk of HCC for women. A significant source of IL-6 appears to be the tumor-associated macrophages [92], although HCC cells could secrete IL-6 in a YAP (yes-associated protein)-dependent manner [93]. YAP have been identified as the transcriptional co-activator of the Hippo pathway, overexpressed in 62% of HCC patients as an oncoprotein to modulate liver size and tumorigenesis [94,95,96]. Hippo signaling is an evolutionarily conserved pathway that controls organ size by regulating cell proliferation, apoptosis, and stem cell self-renewal. To conclude, dysregulation of the Hippo pathway contributes to cancer development.

Inflammation plays a pivotal role in the process of cellular malignant transformation. Persistent and unresolved inflammation within the liver has been demonstrated to stimulate the development of hepatocarcinogenesis [97]. Changes in the cellular microenvironment lead to tumoral shifts over time [98]. Several pro-inflammatory cytokines, such as IL-6, TNF, IL-1β, and anti-inflammatory cytokines, such as TGF-α and TGF-β, and also different transcription factors, such as STAT-3 and NF-κβ, together with their signaling pathways, are involved in modulating the development of HCC [97,98,99]. These cytokines are produced by immune cells such as Th1 and Th2 and are triggered by different danger-associated molecular patterns secreted by injured hepatocytes such as high-mobility group box one and S100, IL-1α [98,100]. Reactive oxygen species, such as peroxides, superoxide, hydroxyl radicals, and nitrogen compounds derived from nitric oxide radicals, are generated as reactions to various stressors and inflammations linked to cancer [101] and contribute to the progression of HCC [98,101,102]. It has been shown that the early stages of HCC present an altered immune response regarding stress-related factors, such as the nitrosative stress profile in monocytes and glutathione levels in monocytes and lymphocytes [101] (Figure 4). 

## 5. The Two Faces of IL-6 in HCC Therapies

IL-6 has a dual role in the immune response. It has been described as anti-inflammatory in some cases, whereas in others, it promotes inflammation and immunity. Initially, it has been viewed as a factor of malignancy, with the IL-6/JAK/STAT3 signaling path having effects such as anti-apoptosis, angiogenesis, and proliferation [2], while more recent findings have shown that IL-6 also has beneficial roles regarding anti-tumor immunity.

IL-6 is synthesized by various immune cells, including macrophages, T cells, B cells, and dendritic cells. Despite its initial identification as a B cell growth factor, IL-6 delivers survival and proliferation cues to a wide array of leukocytes. Nevertheless, the signaling of IL-6 is tightly controlled, primarily due to the restricted presence of the IL-6Rα subunit on specific cell types [103]. As an illustration, IL-6 exhibits elevated expression in naive T cells, effectively enhancing T cell survival and minimizing apoptosis. This effect is achieved by downregulating Fas and FasL, which are transmembrane proteins associated with cell death [104]. Moreover, IL-6 overexpression in mice has been correlated with significantly increased numbers of peripheral T cells, which still had a naive phenotype and did not express memory or activation markers, supporting the idea that the enhanced numbers were due to the increased survival of T cells and not the expansion of activated T cells [105].

Recent studies have revealed that IL-6 also influences the enhancement of adhesive characteristics in blood vessels. This impact extends to structures such as high endothelial venules, which facilitate the entry of naive and memory T cells into lymph nodes and the endothelium of blood vessels within tumors. This augmentation of endothelial adhesiveness results in heightened immune cell trafficking, thereby contributing to the immune system’s ability to target cancer cells within the tumor microenvironment [103]. Another advantage of elevated IL-6 levels is its capacity to stimulate the proliferation and regeneration of liver progenitor cells, also known as oval cells. This ability can potentially facilitate liver mass and function restoration [106]. IL-6 initiates the production of hepatocyte growth factor (HGF) [107], which, in turn, triggers the expression of SNHG12 and regulates the Wnt/β-catenin signaling pathway, thus promoting hepatocyte proliferation and regeneration [108,109]. Moreover, in an IL-6 dependent mechanism, HGF facilitates the phosphorylation of STAT3, leading to the activation of this pathway [109]. This has dual consequences: it regulates liver cells proliferation with the help of cyclin D1/p21, while also preventing cell death by elevating the levels of different proteins involved in this process, i.e., FLIP, Bcl-2, Bcl-xL, Ref1, and MnSOD [110].

Moreover, IL-6 and another pro-inflammatory cytokine, IL-1β, have been shown to induce fibroblast growth factor 2 (FGF-2) expression in both normal and HCC cells. When IL-1β was involved, FGF-2 stimulation led to increased levels of the membrane-bound major histocompatibility complex class I-related chain A (MICA)—which is an NK cell activation molecule—as well as decreased levels of human leukocyte antigen (HLA) class I—which is an NK cell inhibitor. Thus, IL-6 indirectly led, through FGF-2 stimulation, to increased NK sensitivity against the cancer cells, showing another possible new role for IL-6 in innate immunity against HCC [111].

The influence of IL-6 on T cells carries significant implications. Research has demonstrated that effector T cells play a crucial role not only in innate anti-tumor immunity but also in the effectiveness of conventional cancer treatments such as radiation therapy [112,113]. Another study aimed to counteract the negative impact of regulatory T cells in patients treated with therapeutic cytokine-induced killer cells (CIK cells) by adding IL-6 to the CIK cells culture medium. There was a decrease in the number of T_reg_/CD4(+) and T_reg_/CD3(+) along with an increase in the cytotoxicity of the CIK cells against HCC in vitro, suggesting the potential use of IL-6 in improving cancer immunotherapy [114]. 

The equilibrium shifts in favor of elevated IL-6 levels in the context of thermal HCC ablation. During ablation, tissue surrounding the tumor site experiences heat-induced damage, leading to the release of neoantigens into the bloodstream, which subsequently triggers a systemic response [115,116]. Additional exploration is necessary in order to uncover whether the cytokines influenced by MWA (microwave ablation) treatment might have an impact on cancer progression, whether that impact be beneficial or detrimental.

Although IL-6 plays a beneficial role in the body’s protective response against tumor cells, clinical studies have demonstrated a substantial decline in the efficacy of therapeutic drugs due to their interaction with IL-6. A study by Wong V et al. [117] found that patients with HBV and high serum IL-6 levels had an increased risk of developing HCC.

Likewise, a study conducted by Yu Li and colleagues proposes that IL-6 could potentially serve as a biomarker indicative of response to sorafenib (PubChem CID:216239) treatment among patients with hepatocellular carcinoma (HCC). Sorafenib is a standard therapeutic agent frequently employed in HCC treatment. This research article posits that IL-6 might contribute to the development of resistance against sorafenib therapy [118]. 

Furthermore, IL-6’s interaction with immunotherapy has come to the forefront. The combination therapy of atezolizumab/bevacizumab (Atezo/Bev) has emerged as a primary approach for advanced HCC [119,120]. However, approximately 20% of patients fail to respond. In a recent study, simultaneous measurement of 34 plasma proteins was conducted, revealing that elevated plasma IL-6 levels served as a notable predictor of non-responsiveness to Atezo/Bev treatment. This finding was corroborated by the observation that the IL-6-high group exhibited significantly shorter progression-free survival and overall survival durations compared to the IL-6-low group [121].

Considering the significant involvement of IL-6 in developing and advancing HCC, reducing IL-6 levels has emerged as a potential therapeutic strategy. A number of preclinical and clinical trials have examined the use of IL-6 inhibitors, such as monoclonal antibodies and small molecule inhibitors, in the treatment of HCC. Thus, a study by Johnson D et al. [122] found that an antibody capable of neutralizing IL-6 enhanced the anti-tumor effects of sorafenib. In this study, the authors discovered that sorafenib’s anti-tumor effects were increased in a mouse model of HCC by targeting the IL-6/JAK2/STAT3 signaling pathway with an IL-6 neutralizing antibody, implying that IL-6 may contribute to HCC patients’ resistance to sorafenib therapy. Similarly, a clinical trial by Abou-Alfa et al. [123] investigated the safety and efficacy of cabozantinib (PubChem CID:25102847) in HCC patients previously receiving sorafenib treatment. The authors also explored the potential effect of IL-6 biomarkers with respect to cabozantinib treatment. According to the study’s findings, high levels of IL-6 were linked to a poor response to cabozantinib therapy. Notably, compared to patients with lower baseline levels of IL-6, those with greater baseline levels of IL-6 had an overall lower response rate and shorter progression-free survival. Collectively, the research suggests that IL-6 could potentially serve as a biomarker indicating the response to cabozantinib treatment in patients with hepatocellular carcinoma. However, a comprehensive understanding of IL-6 and other potential biomarkers in the context of cabozantinib therapy for HCC demands further extensive research. Even though IL-6 can be harnessed to manipulate the tumor microenvironment and enhance the body’s immune response, its suppressive effects on standard systemic treatments significantly overshadow the potential benefits of elevated IL-6 levels.

While it primarily promotes tumor growth and progression, IL-6 also exhibits paradoxical effects under specific contexts, often linked to the tumor microenvironment and stage. It stimulates an anti-tumor immune response by activating immune cells, aiding immune surveillance, and targeting pre-cancerous cells. This dichotomy underscores the intricate dynamics of IL-6 in HCC and how it can influence different liver cancer therapies illustrated in Figure 5. 

Recent research has highlighted the increasing importance of the intestinal microbiota in the development of liver disease, indicating that manipulating the bacterial population in the gut could be useful in the prevention and management of HCC [124]. The connection between IL-6 and the intestinal microbiota holds a crucial role in the development of HCC. A bidirectional link exists between the liver and the gut, as they communicate via the biliary tract, portal vein, and systemic circulation, forming the gut–liver axis [125]. Dysbiosis can potentially disrupt this system, resulting in heightened intestinal permeability. Consequently, bacterial components and detrimental agents are released within the gut–liver axis, triggering an immune response that initiates inflammation and prompts an elevated secretion of IL-6, fostering the progression of HCC [126]. Moreover, elevated IL-6 levels aggravate both intestinal permeability and microbiota changes [127], creating a harmful cycle that contributes to the progression of HCC, as illustrated in Figure 6.

Cancer stem cells (CSCs), also known as tumor-initiating cells, play a crucial role in tumor initiation, growth, recurrence, metastasis, and resistance to cancer therapies. Studies focused on human HCC observed that within mice HCC, there were cells expressing stem cell markers that surprisingly exhibited a loss of TBRII (TGF-β-receptor type II) [128]. TBRII plays a crucial role in mediating cellular responses to TGF-β, maintaining normal cellular homeostasis and tissue development [129]. Aberrant TGF-β signaling can promote tumor growth, invasion, and metastasis in HCC. The tumors’ expression analysis showed a prominent activation of genes related to the IL-6 signaling pathway, including IL-6 and STAT3. This observation suggests that hepatocellular carcinoma might develop from a stem cell that has transformed due to IL-6 stimulation while concurrently deactivating its TGF-β signaling [128]. One study regarding the resection of HCC revealed that CSCs increased immediately after surgical manipulation, and the laparoscopy group released fewer tumor cells into the blood stream. Post-surgery, both IL-6 and IL-8 levels rose, yet the laparoscopic group showed notably smaller mean increases in serum IL-6 and IL-8 levels compared to the open surgery group. This reduction was linked to decreased cancer progression [130]. A separate study also highlighted the adverse consequences of heightened IL-6 levels. The expression of both IL-6 and IL-6R within the hepatocellular carcinoma (HCC) microenvironment was found to contribute to the recurrence of tumors following surgery. This observation suggests that these molecules could serve as potential indicators of recurrence, and might be viable targets for therapeutic intervention, aiming to improve the long-term survival of patients [131].

Additional research endeavors have focused on investigating the influence of cancer-associated fibroblasts (CAFs) on the advancement of HCC [132]. The findings revealed that CAFs play a significant role by secreting elevated levels of IL-6, promoting cancer growth and progression [83]. Consequent activation of Notch signaling through STAT3 Tyr705 phosphorylation promotes stem cell-like characteristics in HCC cells [133,134]. Thus, the size of xenografted liver tumors was significantly larger in nude mice that received a combination of HCC cells and CAFs than those injected with HCC cells alone [132]. These findings indicate that the secretion of IL-6 by CAFs enhances STAT3/Notch signaling, thereby promoting stem cell-like properties in HCC cells.

## 6. IL-6 Targets

Looking towards bioactive compounds derived from plants that target IL-6 in HCC, including *Curcuma phaeocaulis* [135], *Curcuma aromatica*, *Polygonum cuspidatum* [136], and *Silybum marianum* [137], it has been shown that curcumin inhibits IL-6-induced JAK/STAT3 signaling cascades through negative feedback, suppressing the development of inflammation in HCC [135]. Even though their molecular mechanism is not fully understood, both silymarin and resveratrol are also used as alternative treatments for their anti-HCC-associated effects, possibly through the induction of cell cycle arrest [137] and the installation of apoptotic cell death [136].

Therapies directed at targeting interleukin-6 have been observed to enhance outcomes across various types of cancer. Monoclonal antibody (mAbs) use has been suggested to have a positive effect against the development of ovarian cancer [138], renal cancer [139], multiple myeloma [140], prostate cancer [141], non-small cell lung cancer [142], human oral small squamous cell carcinoma [143], and cancer cachexia [144]. Bevacizumab is a humanized recombinant anti-VEGF-A mAbs. It has been shown to have positive effects on the development of HCC in preclinical studies on mice models [145]. In the clinical setting, it has been evaluated in the treatment of HCC both as a single agent [146], and in combination with immune check point inhibitors such as Atezolizumab [119,120]. Tocilizumab (PubChem SID:472385906) is an anti-IL-6R mAbs approved by the FDA for human use. Its effects have been studied on tumor cell cultures and in mice models. It has been found to inhibit the progression of HCC by reducing the development of cancer stem cells mediated by tumor-associated macrophages [92]. Soluble gp130 (sgp130) is a protein that inhibits the trans signaling pathway of IL-6. In vitro examinations demonstrated that recombinant human sgp130 reduces induced HCC in mice and inhibits the growth and metastases of human HCC xenografts in mice models [147]. 

Evaluation of small-molecule agents that inhibit IL-6 and its associated signaling pathway has been undertaken, exploring a potential approach in cancer treatment [148,149]. Azaspirane (Antiprimod) (PubChem CID:129869) is a JAK2 and JAK3 inhibitor [150], i.e., of enzymes which are involved in the IL-6 signaling pathway [149]. It showed promising results in the in vitro progression of leukemia cells [150]. Antiprimod has been shown to selectively induce apoptosis and inhibit the proliferation of HCC cells that express HBVs or HCVs. This has been demonstrated on HCC culture cells. The mechanisms by which it affects HCC are the deactivation of the Akt and STAT3 signaling pathways. It has also been hypothesized that the accumulation of viral proteins could sensitize cells to antiprimod [151,152].

More recent research has underlined the possible use of newly synthesized azaspirane derivatives, of which 2-(1-(4-(2-cyanophenyl)1-benzyl-1H-indol-3-yl)-5-(4-methoxy-phenyl)-1-oxa-3-azaspiro(5,5) undecane (CIMO) (PubChem CID:14257) was the most promising; CIMO inhibits constitutive and IL-6-induced activation of STAT3, and its inhibitory effect is specific to Tyr-705 in HCC cells, a site of STAT3 phosphorylation, making it a possible treatment for HCC [153].

Another small-molecule compound that inhibits JAKs 1 and 2 is ruxolitinib (PubChem CID:25126798), an FDA-approved drug for treating myelofibrosis (MF) [154]. Ruxolitinib has been shown to have significant anti-proliferative effects on HCC cells, albeit at much higher doses than those approved for MF. It effectively inhibits the downstream targets of JAKs, reducing HCC cell proliferation and colony formation. However, it did not present any in vitro hepatotoxicity, which is a potential risk to be investigated in a clinical setting [155]. Moreover, ruxolitinib has shown positive effects on HCC metastasis progression [156] but has been found to have a higher affinity for specific JAK mutations, such as JAK1-S703I [157]. Furthermore, it was noted that this compound could reduce the progression of hepatic fibrosis and even reverse it [158].

## 7. Conclusions

In conclusion, interleukin-6 assumes a pivotal role in hepatocellular carcinoma, the predominant liver cancer type. Its generation spans across immune, stromal, and cancer cells. Given HCC’s multifaceted nature, effective therapeutic approaches must encompass a range of cellular and molecular targets. This variability in HCC composition implies that elevated IL-6 levels can impact therapies diversely. While IL-6-targeted treatments offer potential, their effectiveness may be augmented by combining them with other modalities, such as immune checkpoint inhibitors or agents targeting angiogenic pathways.

## Figures and Tables

**Figure 1 biomedicines-11-02623-f001:**
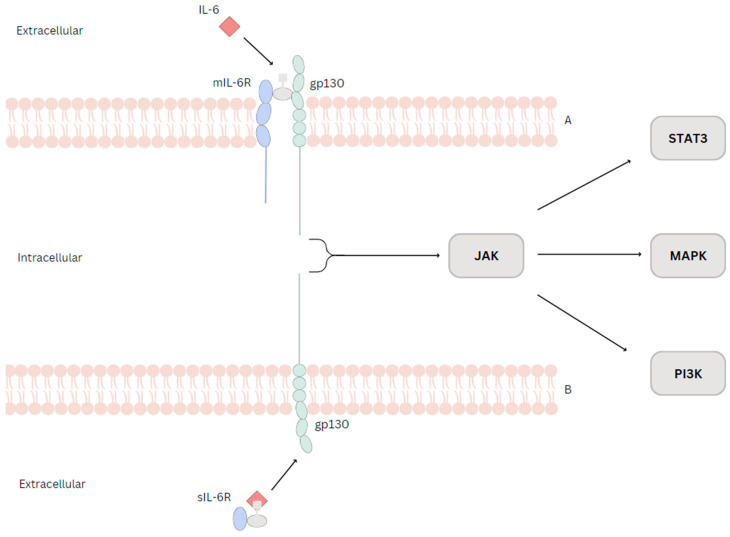
(**A**). Classic IL-6 signaling: IL-6 binds with the cytokine-binding domain of the extracellular portion of mIL-6R, signaling gp130 to activate Janus kinase (JAK). (**B**). Trans IL-6 signaling: the cleaved extracellular portion of mIL-6 and the cytokine-binding domain form the sIL-6R. The complex formed by sIL-6R and IL-6 will signal gp130 to activate Janus kinase (JAK).

**Figure 2 biomedicines-11-02623-f002:**
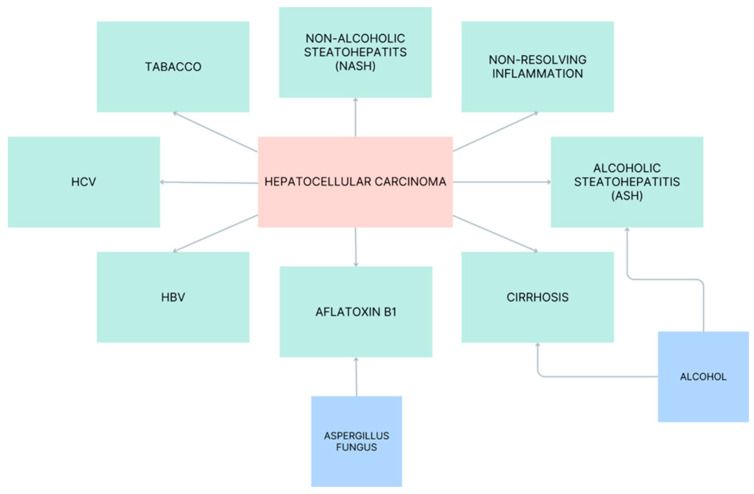
Risk factors for HCC.

**Figure 3 biomedicines-11-02623-f003:**
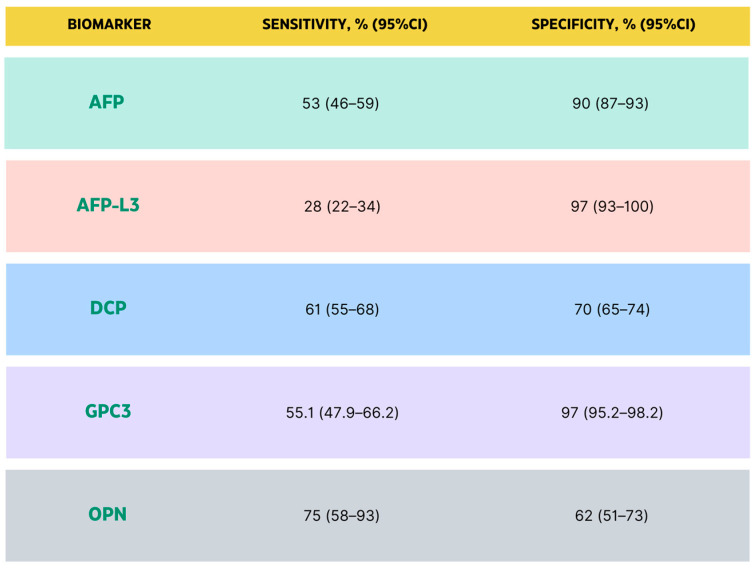
Sensitivity and specificity for AFP [51], AFP-L3 [51], DCP [52], GPC3, and OPN [46].

**Figure 4 biomedicines-11-02623-f004:**
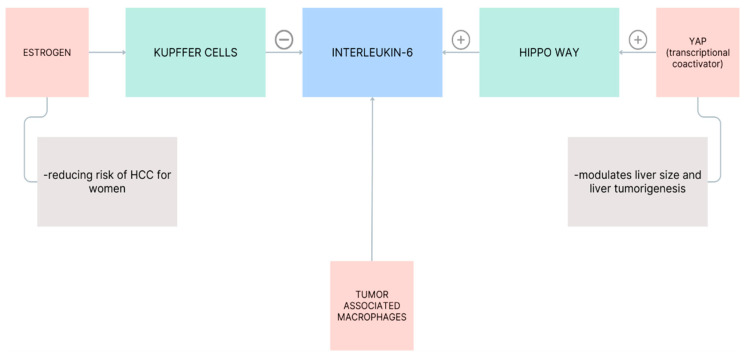
The roles of IL-6 in HCC.

**Figure 5 biomedicines-11-02623-f005:**
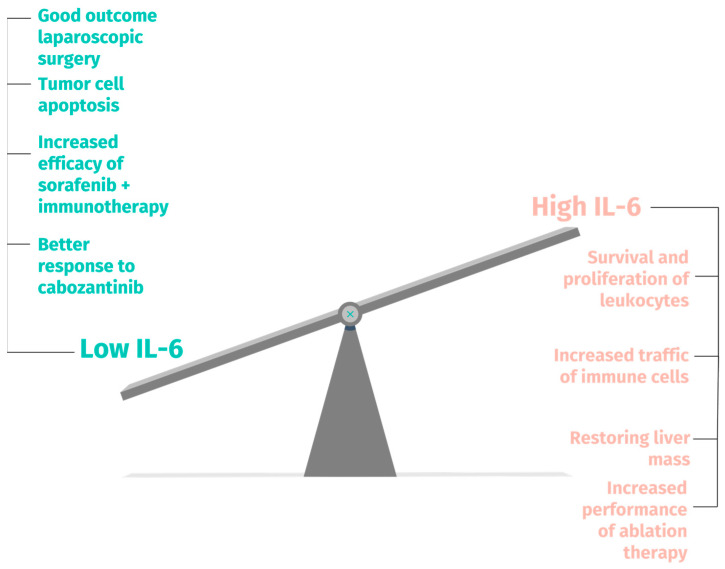
The tipping scale of IL-6.

**Figure 6 biomedicines-11-02623-f006:**
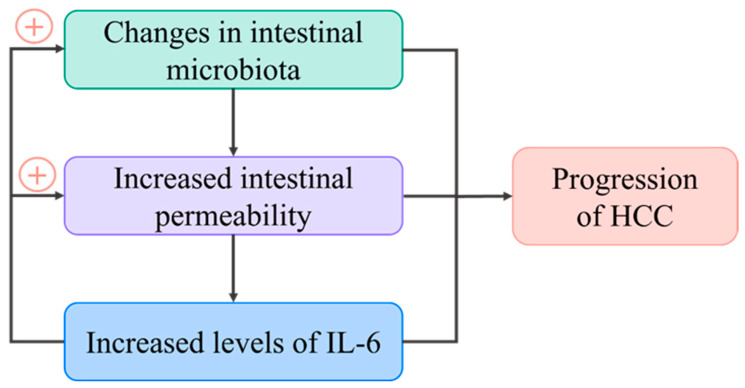
The effect of IL-6 and the intestinal microbiota on the progression of HCC: IL-6 forms a positive feedback loop that perpetuates dysregulation of the gut–immune system axis.

## Data Availability

No new data were created or analyzed in this study. Data sharing is not applicable to this article.

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
