# Peer review of "Interleukin-6 in Hepatocellular Carcinoma: A Dualistic Point of View"

_biomedicines, 2023, doi:10.3390/biomedicines11102623_

Round 1
Reviewer 1 Report
The review article by Nenu et al. intends to summarize the functional role of IL-6 in hepatocellular carcinoma, the use as biomarker and as therapeutic target. The cytokine plays a role in several cancer types and it would be important to highlight, to which extent the actions is similar to other cancers. The role in liver generation could be described in more detail. While the authors cite sensitivity and specificity of the established markers, they do not discuss, if Il-6 has the chance to be a useful biomarker knowing that it is involved in multiple processes.
Minor
l.75: is a part of the sentence missing
l.76 Interleukin 6 written twice
l. 276: abbreviation NASH was already introduced.
English language is fine.
Author Response
REVIEWER 1
- The cytokine plays a role in several cancer types and it would be important to highlight, to which extent the action is similar to other cancers.
- We sincerely thank the reviewer for their valuable perspective, and we will certainly highlight the extent of cytokine action in different cancer types. We have made the following correction:
- IL-6 has been identified as a cytokine that is significantly abundant within the tumor mi-croenvironment (TME) across a spectrum of cancer types. These include head and neck squamous cell carcinoma (HNSCC), pancreatic cancer, non-small-cell lung cancer, breast cancer, ovarian cancer, and melanoma, as substantiated by previous studies [57–67]. Generally, the overexpression of IL-6 is associated with poor prognosis and low survival rate in patients with breast cancer [68]. Beyond its recognized role in tumorigenesis, IL-6 also plays a pivotal role in orchestrating the intricate sequence of events that precede the establishment of secondary tumors, commonly referred to as metastasis.
- The role in liver generation could be described in more detail.
- We would like to express our gratitude to the reviewer for their valuable input, and we will indeed expand upon the description of cytokine roles in liver regeneration as suggested. We have made the following correction:
- IL-6 initiates the production of hepatocyte growth factor (HGF) [105], which in turn trig-gers expression of SNHG12 and regulates the Wnt/β-catenin signaling pathway, thus promoting hepatocyte proliferation and regeneration [106,107]. Moreover, in an IL-6 de-pendent mechanism, HGF facilitates the phosphorylation of STAT3, leading to the activa-tion of this pathway [107]. This has dual consequences: it regulates liver cells proliferation with the help of cyclin D1/p21, while also preventing cell death by elevating levels of dif-ferent proteins involved in this process: FLIP, Bcl-2, Bcl-xL, Ref1, and MnSOD [108].
- While authors cites sensitivity and specificity of the established markers, they do not discuss, if IL-6 has the chance to be a useful biomarker knowing that it is involved in multiple processes.
- We sincerely thank the reviewer for highlighting the need to discuss the potential utility of IL-6 as a biomarker, considering its involvement in various processes.We have made the following correction:
- A study by Gosain et. al aimed to establish the role of IL-6 as a prognostic biomarker in patients with both hepatocellular carcinoma (HCC) and biliary cancer. They assessed how IL-6 impacts two crucial factors affecting quality of life: pain scores and performance status.
In order to minimize potential confounding variables, the study researchers selected a control group comprising 91 individuals. These controls were carefully matched in terms of age, gender, and BMI, as these factors are widely recognized for their substantial impact on IL-6 levels (1–6).The research team collected blood samples from the patients, and serum specimens that were determined using a commercially available enzyme-linked immunosorbent assay (ELISA) kit. IL-6 levels were notably higher in hepatobiliary cancer patients when compared to the healthy control group.
In conclusion, elevated levels of IL-6 were closely associated with poorer prognoses, increased pain scores, and deteriorated performance statuses in the patients (7).
A retrospective study by Yang et al. aimed to assess the utility of IL-6 promoter methylation levels as a noninvasive diagnostic biomarker for liver cancer. The study involved 165 patients diagnosed HCC associated with hepatitis B virus, 198 patients with chronic hepatitis B, and 31 healthy individuals serving as controls. The research employed Methylight methodology to detect methylation levels in the IL-6 promoter region within peripheral blood mononuclear cells (PBMCs). (8)
In conclusion, the study revealed that in HBV-associated HCC, the methylation levels of the IL-6 promoter were notably lower than those observed in patients with chronic hepatitis B. Conversely, the IL-6 mRNA levels were significantly elevated in HBV-associated HCC compared to both CHB patients and healthy individuals. (8)
- MINOR: I.75: Is a part of the sentance missing; I 76 Interleukin 6 written twice; I 276 : abbreviation NASH was already introduced
- The authors have made the modifications
Reviewer 2 Report
In the manuscript "Interleukin-6 in Hepatocellular Carcinoma: A Dualistic Perspective" by Iuliana Nenu and colleagues. It has been reported that hepatocellular carcinoma (HCC) is a pressing health problem that requires an understanding of various mediators' roles in its development in order for therapeutic progress to be made. Because of its intricate and context-dependent functions, Interleukin-6 (IL-6) has taken center stage in research. The purpose of this review is to explore the dual role of IL-6 in HCC, in which it appears to play both a pro- and an inhibitory role in the progression of the disease. A discussion of the protumorigenic effects of IL-6 is presented, including its effects on tumor growth, angiogenesis, and metastasis. In addition, we examine its anti-tumorigenic properties, including its role in immune response activation, induction of cellular senescence, and surveillance of tumors. The purpose of this review is to explore in detail the intricate interactions between IL-6 and the tumor microenvironment, emphasizing the need for a more nuanced understanding of IL-6 signaling in HCC. Accordingly, tailored therapeutic strategies are necessary that take into account the dynamic stages and diverse environments present within the tumor microenvironment. Developing more effective treatment strategies and improving patient outcomes will be made possible by unraveling the multifaceted mechanisms of IL-6 in HCC through future research directions. Regarding the manuscript, I would like to make a few comments.
-The authors should follow the author's instructions regarding the manuscript style and paragraph spacing.
-It is unclear why Section 1 has been introduced, why it is evaluated as dual role, which is the current significance, the authors are required to include information about HCC as well as when Il-6 levels are higher or lower in clinical manifestations
-A number of figures have been created by the authors, perhaps they should summarise or combine some of them into panels.
-Perhaps section 4 required more information, since it shows IL-6 in HCC
-What is the purpose of the introduction of microbiota in this paper?
-Also, Section 6 requires additional information and should be moved to the beginning of the manuscript.
Author Response
We have attached a word document with our replies and changes.
REVIEWER 2
- The authors should follow the author’s instructions regarding the manuscripts style and paragraphs spacing
- The authors have made the modifications.
- It is unclear why Section 1 has been introduced, why it is evaluated as dual role, which is the current significance, the authors are requiered to include information about HCC as well as when IL-6 lever are higher or lower in clinical manifestations
- The authors have made a few changes to Section 1 in order to highlight the significance of the dual role. Moreover, the levels of IL-6 regarding clinical manifestations have been presented in Section 5.
- A number of figures have been created by the authors, perhaps they should summarise or combine some of them into panels
- We greatly appreciate the reviewer's thoughtful suggestion regarding the figures in our manuscript. Each figure has been carefully crafted to align with its respective section and combining them may compromise their individual relevance and clarity. Nevertheless, we will revisit the figures with your input in mind to ensure optimal presentation. Thank you for your valuable input on this matter.
- Perhaps section 4 required more information, since it shows IL-6 in HCC
- IL-6 has been identified as a cytokine that is significantly abundant within the tumor mi-croenvironment (TME) across a spectrum of cancer types. These include head and neck squamous cell carcinoma (HNSCC), pancreatic cancer, non-small-cell lung cancer, breast cancer, ovarian cancer, and melanoma, as substantiated by previous studies [57–67].
Generally, the overexpression of IL-6 is associated with poor prognosis and low survival rate in patients with breast cancer [68]. Beyond its recognized role in tumorigenesis, IL-6 also plays a pivotal role in orchestrating the intricate sequence of events that precede the establishment of secondary tumors, commonly referred to as metastasis [69].
- The roles of IL-6 in HCC are emphasized by the effects of the IL-6/STAT3 signaling pathway. The hyperactivation of STAT3 holds a significant role in the formation of in-flammatory tumors within the microenvironment, facilitating both uncontrolled prolifera-tion of cancer cells and metastasis [73]. Particulary, the genes responsible for promoting cancer cell proliferation, including Ras, Src, and cyclin D1, serve as direct targets of STAT3 [74].
Tumor-associated macrophages (TAMs) promote tumor progression in liver tissue microenvironments by releasing IL-6, which in turn activates the IL-6/STAT3 signaling pathway in nearby HCC stem cells [75]. A study by Zheng et al. [76] sheds light on the hyperactivation of the IL-6/STAT3 signaling pathway in HCC cells which could upregu-late the tissue inhibitor of metalloproteinases-1 (TIMP-1), thus prompting the conversion of normal liver fibroblasts (LFs) into carcinoma-associated fibroblasts (CAFs), ultimately driving the initiation of liver cancer [77].
The induction of apoptosis in HCC cells primarily hinges on two mechanisms: the upregulation of anti-apoptotic factors and the promotion of survival signals. Following STAT3 activation mediated by IL-6, there is an important increase in the expression of an-ti-apoptotic proteins such as Bcl-xL, Bcl-2, survivin, and P53, among others. This upregu-lation plays a crucial role in safeguarding HCC cells against apoptosis [76,78–80]. Partic-ulary, Bcl-2 stands out as a vital protein in fostering the survival of tumor cells.
The critical determinant of apoptosis is the delicate balance between pro-apoptotic and anti-apoptotic proteins. Activation of the IL-6/STAT3 signaling pathway has the po-tential to shift this balance by increasing the proportion of pro-apoptotic factors relative to anti-apoptotic factors, and heightened IL-6 levels are likely to contribute to this modifica-tion [81].
Additionally, STAT3 phosphorylation can directly bind to the promoter region of the survivin gene, leading to an upregulation of survivin expression and thereby promoting the survival of tumor cells. On the other hand, inhibiting STAT3 activity can result in the downregulation of survivin gene expression, facilitating the induction of apoptosis in liver cancer cells [82]. These findings demonstrate that the activation of the IL-6/STAT3 signal-ing pathway has the capacity to boost the expression of survival-related proteins, which in turn act to impede apoptosis in HCC cells.
A crucial role in tumor invasion and metastasis is vascular endothelial growth factor (VEGF) which promotes vascular endothelial cell growth and tumor neoangiogenesis. The expression of VEGF is more elevated in liver tumor tissue than in cirrhosis and normal liver tissue [83]. A significant factor which can induce the secretion and expression of VEGF in tumor tissue is hypoxia through hypoxia inducible factor 1 (HIF-1). IL-6 binds with IL-6R to induce the activation of STAT3 and activated STAT3 binds to the promoter region of the VEGF gene to increase transcription, promoting the tumor angiogenesis.
- What is the purpose of the introduction of microbiota in this paper?
- We would like to express our gratitude to the reviewer for raising this question. In our paper, we introduced the topic of microbiota to underscore its growing significance in recent research, particularly its role in the development of liver diseases, including hepatocellular cancer, and how this relates to our discussion of IL-6. Your input has allowed us to clarify the purpose of including microbiota in our introduction. Thank you for your valuable insight on this matter.
- Section 6 requires additional information and should be moved to the beginning of the manuscript
- To increase the relevance of Section 6, it is crucial to include supplementary information about IL-6, specifically detailing its roles(Section 4) and mechanisms(Section 2).
- Bevacizumab is a humanized recombinant anti-VEGF-A mAbs. It has been shown to have positive effects on the development of HCC in preclinical studies on mice models [143]. In the clinical setting, it has been evaluated in the treatment of HCC both as a single agent [144], and in combination with immune check piont inhibitors such as Atezolizumab [117]. Tocilizumab is an anti-IL-6R mAbs approved by the FDA for human use. Its effects have been studied on tumor cell cultures and in mice models. It has been found to inhibit the progression of HCC, by reducing the development of cancer stem cells mediated by tumor associated macrophage [145]. Soluble gp130 (sgp130) is a protein that inhibits the trans signaling pathway of IL-6. In vitro examinations demonstrated that recombinant human sgp130 reduce induced HCC in mice and inhibits growth and metastases of hu-man HCC xenografts in mice models [146].
Evaluation of small molecule agents that inhibit IL-6 and its associated signaling pathway has been undertaken as a potential approach in cancer treatment [147,148]. Azaspirane (Antiprimod) is an JAK2 and JAK3 inhibitor [149], enzymes which are in-volved in IL-6 signaling pathway [148]. It showed promising results in the in vitro pro-gression of leukemia cells [149]. Antiprimod has been shown to selectively induce apop-tosis and inhibit the proliferation of HCC cells that expressed HBVs or HCVs. This has been demonstrated on HCC culture cells. The mechanisms by which it affects HCC are the deactivation of the Akt and STAT3 signaling pathways. It has also been hypothesized that the accumulation of viral proteins could sensitize cells to antiprimod [120]
Reviewer 3 Report
Hepatocellular Carcinoma (HCC) is a pressing health concern, demanding a deep understanding of various mediators' roles in its development for therapeutic progress. Notably, Interleukin-6 (IL-6) has taken center stage in investigations due to its intricate and context-dependent functions. This review delves into the dual nature of IL-6 in HCC, exploring its seemingly contradictory roles as both a promoter and an inhibitor of disease progression.
Given HCC multifaceted nature, effective therapeutic approaches must encompass a range of cellular and molecular targets. This variability in HCC composition implies that elevated IL-6 levels can impact therapies diversely. While IL-6-targeted treatments offer potential, their effectiveness may be augmented by combining them with other modalities, like immune checkpoint inhibitors or agents targeting angiogenic pathways.
The review work provides a possible vision of the future of treatment for the treatment of hepatocellular carcinoma. This is its strong point. On the other hand, the bibliographic references should be updated, providing some from this year. Furthermore, in the references part, sometimes the authors are written with all letters in capital letters and sometimes not. Uniformity must be maintained.
Drugs are named in the text; It is necessary to provide the corresponding chemical structures to facilitate the understanding of chemistry.
The level of English is correct.
Author Response
We have attached a word document with our replies and changes.
Reviewer 3
- The bibliographic references should be updated, providing some from this year?
- Thank you for your valuable feedback regarding the bibliographic references in our manuscript. We understand the importance of keeping the references up-to-date to ensure the highest quality and relevance of our work. In response to your comment, we have carefully reviewed and updated the references in our manuscript.
- Rose-John, S.; Jenkins, B.J.; Garbers, C.; Moll, J.M.; Scheller, J. Targeting IL-6 Trans-Signalling: Past, Present and Future Prospects. Nat Rev Immunol 2023, doi:10.1038/s41577-023-00856-y.
- Yıldırım, H.Ç.; Kavgaci, G.; Chalabiyev, E.; Dizdar, O. Advances in the Early Detection of Hepatobiliary Cancers. Cancers (Basel) 2023, 15, doi:10.3390/cancers15153880.
- Tayob, N.; Kanwal, F.; Alsarraj, A.; Hernaez, R.; El-Serag, H.B. The Performance of AFP, AFP-3, DCP as Biomarkers for Detection of Hepatocellular Carcinoma (HCC): A Phase 3 Biomarker Study in the United States. Clinical Gastroenterology and Hepatology 2023, 21, 415-423.e4, doi:10.1016/j.cgh.2022.01.047.
- Makino, Y.; Hikita, H.; Kato, S.; Sugiyama, M.; Shigekawa, M.; Sakamoto, T.; Sasaki, Y.; Murai, K.; Sakane, S.; Kodama, T.; et al. STAT3 Is Activated by CTGF-Mediated Tumor-Stroma Cross Talk to Promote HCC Progression. Cell Mol Gastroenterol Hepatol 2023, 15, 99–119, doi:10.1016/j.jcmgh.2022.09.006.
- Yang, H.; Kang, B.; Ha, Y.; Lee, S.H.; Kim, I.; Kim, H.; Lee, W.S.; Kim, G.; Jung, S.; Rha, S.Y.; et al. High Serum IL-6 Correlates with Reduced Clinical Benefit of Atezolizumab and Bevacizumab in Unresectable Hepatocellular Carcinoma. JHEP Reports 2023, 5, 100672, doi:10.1016/j.jhepr.2023.100672.
- Gajos-Michniewicz, A.; Czyz, M. WNT/β-Catenin Signaling in Hepatocellular Carcinoma: The Aberrant Activation, Pathogenic Roles, and Therapeutic Opportunities. Genes Dis 2024, 11, 727–746, doi:10.1016/j.gendis.2023.02.050.
- In the references part, sometimes the authors are written with all letters in capital letters and sometimes not
- Thank you for your review, and we have now rectified the capitalization inconsistency in the author names within the references section
- Drugs are named in the text; It is necessary to provide the corresponding chemical structures to facilitate the understanding of chemistry.
- We sincerely appreciate your suggestion to include chemical structures for the drugs mentioned in our manuscript. We have now incorporated these structures to enhance the understanding of the chemistry involved, and we thank you for your valuable input.
- sorafenib (PubChem CID:216239)
- cabozantinib (PubChem CID:25102847)
- Tocilizumab(PubChem SID:472385906)
- Azaspirane (Antiprimod) (PubChem CID:129869)
- undecane (CIMO) (PubChem CID:14257)
- ruxolitinib (PubChem CID:25126798)
Round 2
Reviewer 2 Report
Thank you for taking into account my previous comments. As a result, the manuscript now reads well and the previous questions have been adequately addressed.
Reviewer 3 Report
The work is well constructed and with the new comments introduced, it has gained scientific quality. On the other hand, an access route to organic compounds is shown that in some way alleviates the fact that chemical structures are not drawn. Probably, the authors do not have programs that allow them to do this task.
In conclusion, the work in its current form can be accepted for publication.